# An AI-Based Exercise Prescription Recommendation System

**Hung-Kai Chen [1],\*, Fueng-Ho Chen [2] and Shien-Fong Lin [3]**

1   Institute of Electrical and Computer Engineering, College of Electrical and Computer Engineering, National Chiao Tung University, 1001 University Road, Hsinchu 30010, Taiwan
2   JoiiUp Technology Corporation, Hsinchu 30264, Taiwan; herman_chen@joiiup.com
3   Institute of Biomedical Engineering, College of Electrical and Computer Engineering, National Chiao Tung University, 1001 University Road, Hsinchu 30010, Taiwan; linsf5402@nctu.edu.tw
\*   Correspondence: jackchen6689.cm05g@nctu.edu.tw

**Abstract:** The European Association of Preventive Cardiology Exercise Prescription in Everyday Practice and Rehabilitative Training (EXPERT) tool has been developed for digital training and decision support in cardiovascular disease patients in clinical practice. Exercise prescription recommendation systems for sub-healthy people are essential to enhance this dominant group's physical ability as well. This study aims to construct a guided exercise prescription system for sub-healthy groups using exercise community data to train an AI model. The system consists of six modules, including three-month suggested exercise mode (3m-SEM), predicted value of rest heart rate (rest HR) difference after following three-month suggested exercise mode (3m-PV), two-month suggested exercise mode (2m-SEM), predicted value of rest HR difference after following two-month suggested exercise mode (2m-PV), one-month suggested exercise mode (1m-SEM) and predicted value of rest HR difference after following one-month suggested exercise mode (1m-PV). A new user inputs gender, height, weight, age, and current rest HR value, and the above six modules will provide the user with a prescription. A four-layer neural network model is applied to construct the above six modules. The AI-enabled model produced 95.80%, 100.00%, and 95.00% testing accuracy in 1m-SEM, 2m-SEM, and 3m-SEM, respectively. It reached 3.15, 2.89, and 2.75 BPM testing mean absolute error in 1m-PV, 2m-PV, and 3m-PV. The developed system provides quantitative exercise prescriptions to guide the sub-healthy group to engage in effective exercise programs.

**Keywords:** exercise prescription; suggested exercise mode; rest heart rate

## 1. Introduction

Almost every government in the world spends a lot of effort to fight the increasingly serious problem of chronic diseases. According to a report by the Centers for Disease Control and Prevention, 90% of the US$3.8 trillion in healthcare expenditures per year in the United States is attributable to chronic diseases, including heart disease, stroke, cancer, type 2 diabetes, obesity, and arthritis [1]. Effective exercise can effectively prevent and even treat many chronic diseases. In particular, related chronic diseases caused by factors such as "insufficient exercise" and "obesity" depend on continuous and effective exercise to achieve the effect of prevention or improvement. Engaging in valid and safe exercise is as effective as drugs for the prevention and treatment of cardiovascular disease (CVD), and without side effects. Therefore, taking effective exercise as the prescription and the first line of defense for the prevention of chronic diseases has received extensive support and initiatives. Body composition such as blood pressure, glycaemic and lipid profile improve obviously in patients with CVD risk while participating in exercise training [2–4]. Exercise training is therefore classified as a type 1A intervention in the treatment of CVD risk [5].

In the past few decades, lots of clinical data have been collected across different sites. With the growth of information technology, these data provided a high value of digital information to integrate into the healthcare recommendation system. These systems gave patients a personalized recommendation and improved understanding of their

medical condition. Personalized diets, exercise routines, medications, disease diagnoses, and other healthcare services all belong to the domain of healthcare recommendation systems. In addition to health-related recommendation systems, various recommendation systems have been integrated into online retailers, streaming services, social networks, physical assistants, and e-commerce applications [6–8]. The current AI-driven global health interventions cover four categories relevant to global health researchers: (1) diagnosis, (2) patient morbidity or mortality risk assessment, (3) disease outbreak prediction and surveillance, and (4) health policy and planning [9]. Focus on health policy and planning of previous healthcare recommendation systems, collaborative filtering, content-based, knowledge-based and hybrid approaches are the basic recommendation techniques in health recommender systems [10,11]. The preventive programs for therapy optimization, adherence and risk factor management including exercise training, are now recommended for patients with CVD to reduce disease recurrence by the 2016 European guidelines for CVD prevention [5]. The European Association of Preventive Cardiology developed a digital training and decision support system for optimized exercise prescription in CVD patients based on the definition and diagnostic criteria for diseases and risk factor [12,13].

On the other hand, up to 75% of the world's people are defined as belonging to the "sub-healthy group" according to the statistics from the World Health Organization [14]. The health condition of this group is needed to be take care as well. In addition, high rest heart rate (rest HR) exists higher risk to suffer from CVD such as obesity, diabetes, dyslipidaemia and hypertension than low rest HR [15–19]. Additionally, exercising with a period of time has been proved to decrease rest HR [20,21]. In this paper, we aim to develop an exercise prescription system to provide suggested exercise mode and the predicted value of rest HR difference if the user follows the suggested exercise mode in future three-month period, and hereby to contribute to increase health condition for the sub-healthy group.

## 2. Methods

### 2.1. Data Sources

The construction of exercise prescription system is based on the data from exercise-based social platform Joiisports [22]. Joiisports is a popular sports-based social platform in Taiwan. The platform contains more than 100,000 active users, and about 70% of active users using wearable heart rate devices. The database of JoiiSports is open access for employees and students of National Chiao Tung University. The usage privilege of Joiisports database in this study obtained the written consent by JoiiUp Technology Corporation. Each user's name from database has been de-identified before starting the experiment.

Each single user information data of the database including user ID, gender, height, weight and age. Each single rest HR data of the database including rest HR, measure time and measure date. Each single exercise data of the database including start time, total exercise time, effective exercise time, average exercise HR, maximum HR, minimum HR, type of exercise (running, biking, hiking, mountaineering, aerobic exercise, ball games, indoor sports, step ups, spinning bike, swimming, Pilates, strength training, cardiorespiratory endurance, martial arts, stretching, indoor running, resistance exercise, intermittent exercise and step counting), Metabolic Equivalent of Task (METs), HR intensity zone, total calories, effective exercise calories, exercise distance, exercise efficiency and exercise track. METs can also be regarded as the basis of exercise intensity in each single exercise.

### 2.2. Data Preprocessing

At first, user information data, rest HR data and exercise data are bound together by each user ID. Each month's all rest HR data are averaged to represent the month's rest HR data for each user. Each month's all effective exercise time data and average exercise HR data are averaged to represent the month's effective exercise time data and average exercise HR data. The number of occurrences of each exercise type, accumulated MET $\times$ exercise time in minutes (METs-min) value and exercise frequency in each month are calculated as well. METs-min can also be regarded as the basis of total energy consumption in each single exercise.

The exercise prescription system is consisted of six modules. Three-month suggested exercise mode (3m-SEM), predicted value of rest HR difference after following three-month suggested exercise mode (3m-PV), two-month suggested exercise mode (2m-SEM), predicted value of rest HR difference after following two-month suggested exercise mode (2m-PV), one-month suggested exercise mode (1m-SEM) and predicted value of rest HR difference after following one-month suggested exercise mode (1m-PV) are constructed.

### 2.3. Exercise Scenario

The exercise prescription system provides a three-month exercise mode suggestion to new user. While a new user input gender, height, weight, age and current rest HR value, 3m-SEM and 3m-PV will feedback to user at first. If the user could achieve first month's exercise performance expectation, the user will keep using 3m-SEM in future two month. If the user could achieve second month's exercise performance expectation, the user will keep using 3m-SEM in last month. It represented that the user has followed the suggested exercise mode of 3m-SEM for three-month period. If the user could not achieve first month's exercise performance expectation from 3m-SEM, 2m-SEM and 2m-PV will feedback to user. If the user could achieve second month's exercise performance expectation from 2m-SEM, the user will keep using 2m-SEM in last month. It represented that the user has followed the suggested exercise mode of 2m-SEM for two-month period. If the user could not achieve second month's exercise performance expectation from 2m-SEM, 1m-SEM and 1m-PV will feedback to user. In addition, if the user could not achieve second month's exercise performance expectation from 3m-SEM, 1m-SEM and 1m-PV will feedback to user.

Within 3m-SEM, the suggested effective exercise time, average exercise HR, type of exercise, METs-min and exercise frequency value of exercise mode is averaged by three-month duration. Thus, the built suggested exercise modes in each month are the same inside 3m-SEM. Identically, suggested exercise mode inside 2m-SEM are the same in each month. On top of that, each suggested exercise mode's recommended METs-min value is regarded as the exercise performance criterion. The exercise scenario is demonstrated in Figure 1A.

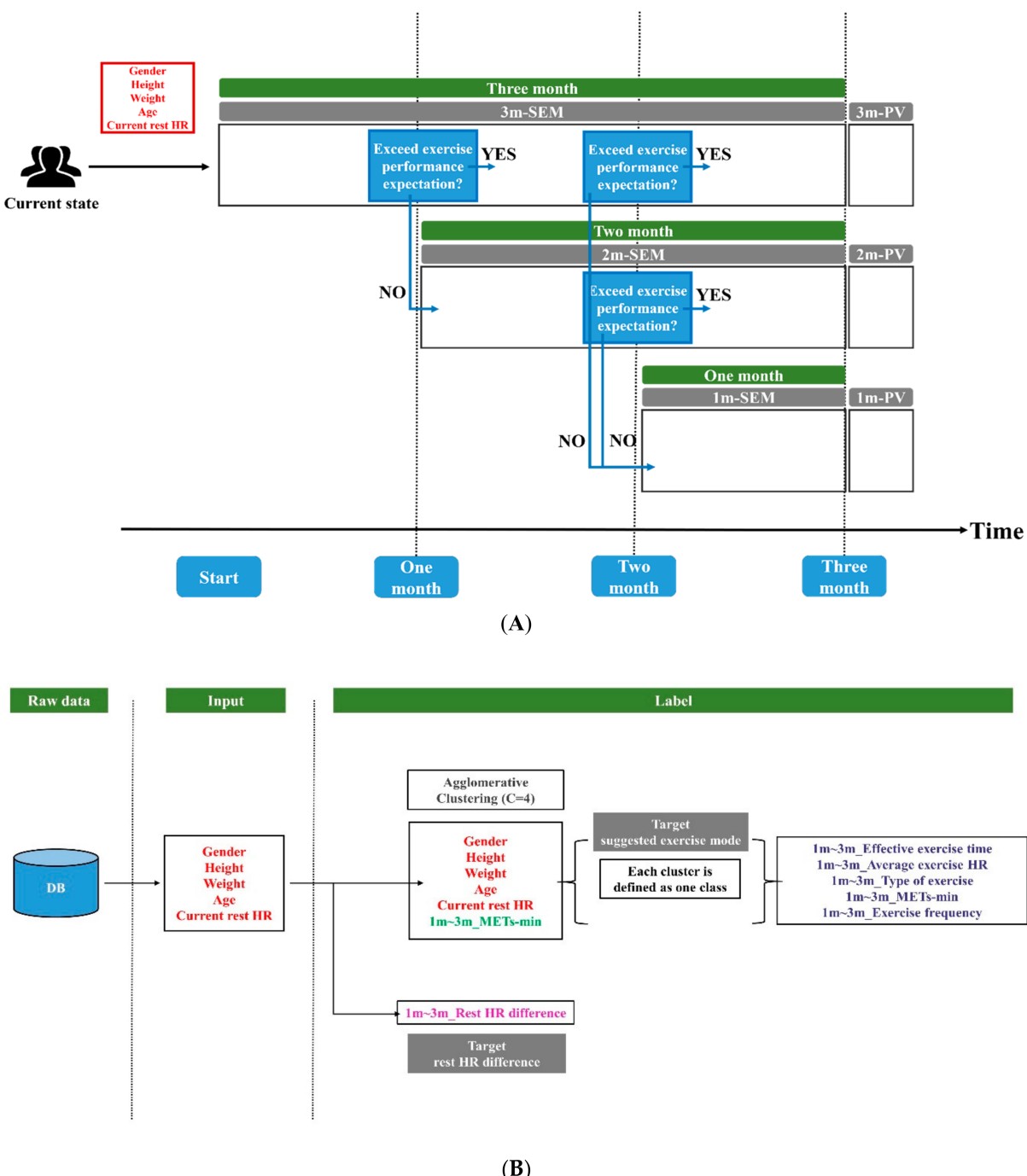

**Figure 1.** (**A**) The exercise scenario of this study; (**B**) The flowchart to define input data and target of SEM modules and PV modules. SEM: suggested exercise mode. PV: predicted value of rest HR difference after following suggested exercise mode.

### 2.4. Modules Construction

The data with rest HR dropped more than four beats per minute (BPM) after one-month exercise are extracted as 1m-SEM and 1m-PV's training data. The data with rest HR dropped more than four BPM after two-month exercise are extracted as 2m-SEM and 2m-PV's training data. Identically, the data with rest HR dropped more than four BPM after three-month exercise are extracted as 3m-SEM and 3m-PV's training data. Taking 3m-SEM for example, we adopted gender, height, weight, age, current rest HR value as input data. While defining target of suggested exercise mode, the above input data and three-month average METs-min are combined at first. Agglomerative clustering [23,24] is applied to

divide different exercise mode by the above six-dimension indices. The amount of cluster is set to 4. Each cluster is labeled as one class. Identically, the amount of cluster is set to 4 for 2m-SEM and 1m-SEM's training as well. The suggested effective exercise time, average exercise HR, type of exercise, METs-min and exercise frequency of 1m-SEM, 2m-SEM and 3m-SEM in each class is demonstrated in the result section. Each mean value of effective exercise time, average exercise HR, type of exercise, METs-min and exercise frequency from the users inside each class are calculated to represent each suggested indices. Gender, height, weight, age, current rest HR value are PV modules' input data as well. The rest HR difference after-one month, two-month and three-month exercise are selected as 1m-PV, 2m-PV and 3m-PV's target value. The flowchart to define the input data and the target of 1m-SEM, 2m-SEM, 3m-SEM, 1m-PV, 2m-PV and 3m-PV is demonstrated in Figure 1B.

We expand the dimensionality of 1m-SEM, 2m-SEM, 3m-SEM, 1m-PV, 2m-PV and 3m-PV's input data by encoding gender, height, weight, age, and current rest HR value to one-hot vectors by different value range. The dimension of the input data augmented from 5 to 24 after one-hot encoding. The range to transfer each dimension's input data values into one-hot vectors are demonstrated in Table 1. METs-min value used for agglomerative clustering are encoded by the range of [0,2000), [2000,3000), [3000,4000), [4000,5000) and [5000,∞) as well.

**Table 1.** Different range to transfer raw value of input data into one-hot vector.

| Raw Data | One-Hot Encoding | Transferred Data |
|---|---|---|
| Gender | 0 | [1,0] |
|  | 1 | [0,1] |
| Height | Height < 150 | [1,0,0,0,0] |
|  | 150 ≦ Height < 160 | [0,1,0,0,0] |
|  | 160 ≦ Height < 170 | [0,0,1,0,0] |
|  | 170 ≦ Height < 180 | [0,0,0,1,0] |
|  | 180 ≦ Height | [0,0,0,0,1] |
| Weight | Weight < 50 | [1,0,0,0,0] |
|  | 50 ≦ Weight < 70 | [0,1,0,0,0] |
|  | 70 ≦ Weight < 90 | [0,0,1,0,0] |
|  | 90 ≦ Weight < 110 | [0,0,0,1,0] |
|  | 110 ≦ Weight | [0,0,0,0,1] |
| Age | Age < 17 | [1,0,0,0,0,0,0] |
|  | 17 ≦ Age < 25 | [0,1,0,0,0,0,0] |
|  | 25 ≦ Age < 35 | [0,0,1,0,0,0,0] |
|  | 35 ≦ Age < 45 | [0,0,0,1,0,0,0] |
|  | 45 ≦ Age < 55 | [0,0,0,0,1,0,0] |
|  | 55 ≦ Age < 65 | [0,0,0,0,0,1,0] |
|  | 65 ≦ Age | [0,0,0,0,0,0,1] |
| Rest HR | Rest HR < 50 | [1,0,0,0,0] |
|  | 50 ≦ Rest HR < 60 | [0,1,0,0,0] |
|  | 60 ≦ Rest HR < 70 | [0,0,1,0,0] |
|  | 70 ≦ Rest HR < 80 | [0,0,0,1,0] |
|  | 80 ≦ Rest HR | [0,0,0,0,1] |

The training and testing procedure of 1m-SEM, 2m-SEM and 3m-SEM are the same. Taking 3m-SEM for example, we divided total data into 4 clusters by agglomerative clustering. Each cluster is labeled as one class. The amount of data from the first group to the fourth group are equal to 60, 103, 58 and 48. The fourth group contains the least amount of data. The 10% amount of data ($n = 5$) inside the fourth group is picked randomly as the fourth group's testing data. After that, the first group, the second group and the third group randomly picked 5 pieces of data respectively to annotated as the rest of testing data. Thus, the total amount of testing data equals to 20. After picking testing data from the total

data, the rest of data is annotated as training data. The total amount of training data of 3m-SEM is equal to 249.

Due to the unbalance problem between each group's data amount, Synthetic Minority Oversampling Technique (SMOTE) algorithm [25] is applied to overcome the problem. After balancing the amount of training data between each group, we picked 6000 pieces of data inside the training data repeatedly to augment the amount of training data. 10-fold cross validation is applied for 3m-SEM's hyper-parameters tuning. Next, we retrained 3m-SEM using fixed hyper-parameters for fine tuning. Testing data is used to test performance of fine-tune 3m-SEM.

A four-layer neural network is applied for 3m-SEM training. The number of nodes in first layer equals to 24 which is corresponding to the dimension of input data. The number of nodes is set to 20 and 10 in the second and the third layer. The last layer is consisted of 4 nodes which is reasoned by the number of classes to predict. The overview of SEM modules' training and testing procedure are shown in Figure 2A. The 1m-SEM, 2m-SEM and 3m-SEM's testing accuracy are shown in the result section.

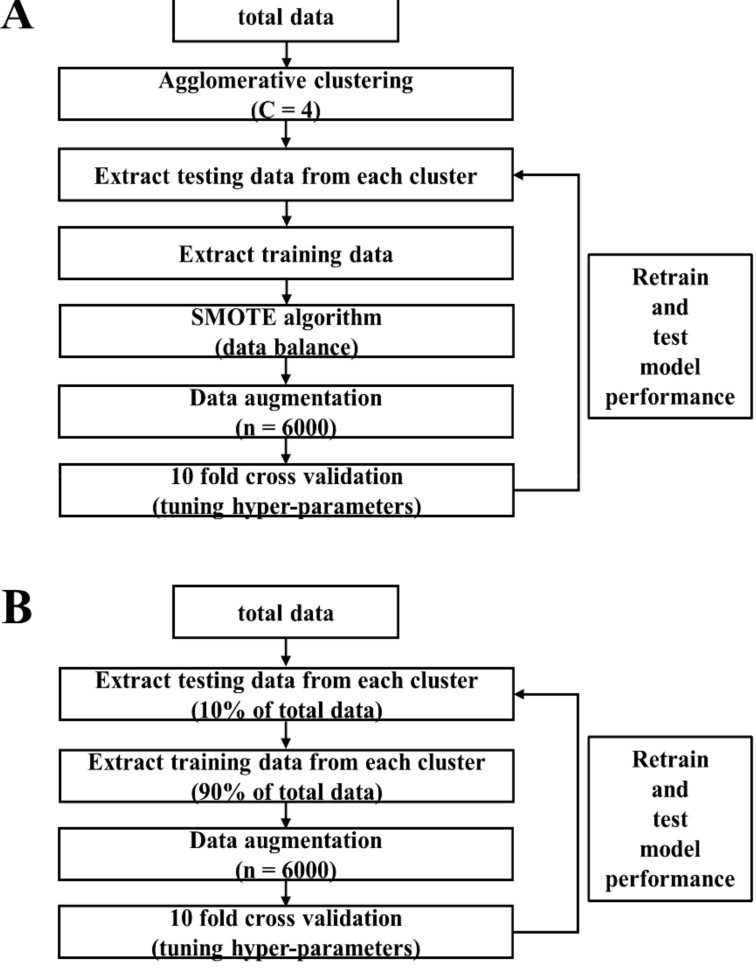

**Figure 2.** (**A**) The overview of SEM modules' training and testing procedure; (**B**) The overview of PV modules' training and testing procedure. SEM: suggested exercise mode. PV: predicted value of rest HR difference after following suggested exercise mode.

The training and testing procedure of 1m-PV, 2m-PV and 3m-PV are the same. Taking 3m-PV for example, we randomly picked 90% of total data as the module's training data. The rest of 10% of total data is used to test module's final prediction accuracy. We picked 6000 pieces of data inside 90% of total data repeatedly in order to augment the amount of training data. 10-fold cross validation is applied for 3m-PV's hyper-parameters tuning.

After that, we retrained 3m-PV using the fixed hyper-parameters for fine tuning. Testing data is used to test performance of fine-tune 3m-PV.

A four-layer neural network is applied to construct 3m-PV. From the first layer to the third layer of 3m-PV, it adopted the same structure as 3m-SEM. The amount of node in the last layer equals to 1 which is reasoned by the single scalar of rest HR difference value we are going to predict. The number of epoch and batch size is fixed to 500 and 16 after hyper-parameters tuning in both SEM and PV modules. Early-stopping function is applied during both 10-fold cross validation and fine tuning. The overview of PV modules' training and testing procedure are shown in Figure 2B. The testing accuracy of 1m-PV, 2m-PV and 3m-PV are shown in the Results section.

## 3. Results

### 3.1. Testing Accuracy of SEM Modules

The 10-fold cross validation accuracy and final testing accuracy of 1m-SEM, 2m-SEM and 3m-SEM are shown in Table 2A. The mean and standard deviation of 10-fold cross validation accuracy from 1m-SEM to 3m-SEM is equal to [95.93% $\pm$ 0.86%, 99.02% $\pm$ 0.34%, 98.45% $\pm$ 0.41%]. The fine-tune 2m-SEM reached 100% testing accuracy. It also reached 95.80% and 95.00% testing accuracy on 1m-SEM and 3m-SEM.

**Table 2.** (**A**). 10-fold cross validation accuracy and fine-tune accuracy of 1m-SEM, 2m-SEM and 3m-SEM. Table 2 (**B**). 10-fold cross validation Mean absolute error (MAE) and fine-tune MAE of 1m-PV, 2m-PV and 3m-PV. Std: standard deviation.

|     | Testing Accuracy (%) | 10-Fold Cross Validation | | Fine-Tune |
| --- | --- | --- | --- | --- |
|     |        | Mean | Std |                |
| (A) | 1m-SEM | 95.93 | 0.86 | 95.80 (46/48)  |
|     | 2m-SEM | 99.02 | 0.34 | 100.00 (28/28) |
|     | 3m-SEM | 98.45 | 0.41 | 95.00 (19/20)  |
|     | **Testing MAE (BPM)** | **10-Fold Cross Validation** | | **Fine-Tune** |
|     |       | Mean | Std |      |
| (B) | 1m-PV | 2.86 | 0.08 | 3.15 |
|     | 2m-PV | 2.72 | 0.09 | 2.89 |
|     | 3m-PV | 2.59 | 0.08 | 2.75 |

### 3.2. Exercise Mode Analysis in Each Group

The suggested effective exercise time, average exercise HR, type of exercise, METs-min and exercise frequency of 1m-SEM in each group are shown in Table 3A. Basically, effective exercise time is required essentially at least 3150 s in each single exercise no matter which group is recommended to a new user. Average exercise HR is required at least 124 in each single exercise no matter which group is recommended to a new user. Running, hiking, and indoor exercise are recommended for a new user inside 1m-SEM. Group 2 and group 4 require more energy consumption relative to group 1 and group 3 followed by METs-min value during one-month exercise. Exercise frequency for at least 3 times a week is recommended to a new user in all of the groups. In addition, it is easier to follow the recommended METs-min value if new user is not easy to control the rest of the indices. For example, if one of the new users is recommended to group 1, achieving METs-min value for at least 7324 in one month is an alternative. It is not necessary to achieve the criterion of effective exercise time, average exercise HR, type of exercise and exercise frequency.

**Table 3.** (**A**). The recommended exercise indices of 1m-SEM in each group; (**B**). The recommended exercise indices of 2m-SEM in each group; (**C**). The recommended exercise indices of 3m-SEM in each group. Unit of effective exercise time: Seconds/each single exercise. Unit of average exercise HR: BPM/each single exercise. Type of exercise: '1' refers to running, '4' refers to hiking, '9' refers to indoor exercise. Unit of Mets-min: (METs × minute)/per month. Unit of exercise frequency: times/each week. G1: group 1. G2: group 2. G3: group 3. G4: group 4.

| | | 1m-SEM ($n = 773$) | | | |
|---|---|---|---|---|---|
| | | G1 ($n = 211$) | G2 ($n = 163$) | G3 ($n = 279$) | G4 ($n = 120$) |
| (A) | Effective exercise time | 3202 | 3249 | 3152 | 3178 |
| | Average exercise HR | 126 | 124 | 128 | 126 |
| | Type of exercise | 1, 4 | 1, 9 | 1, 4 | 1, 4 |
| | Mets-min | 7324 | 8163 | 5922 | 9395 |
| | Exercise frequency | 3.31 | 3.36 | 3.13 | 3.61 |
| | | **2m-SEM ($n = 410$)** | | | |
| | | G1 ($n = 158$) | G2 ($n = 85$) | G3 ($n = 68$) | G4 ($n = 99$) |
| (B) | Effective exercise time | 3109 | 3260 | 3351 | 3269 |
| | Average exercise HR | 127 | 124 | 121 | 128 |
| | Type of exercise | 1, 4 | 1, 4 | 1, 9 | 1, 9 |
| | Mets-min | 7001 | 11,431 | 9245 | 8377 |
| | Exercise frequency | 3.51 | 4.48 | 4.16 | 3.26 |
| | | **3m-SEM ($n = 269$)** | | | |
| | | G1 ($n = 60$) | G2 ($n = 103$) | G3 ($n = 58$) | G4 ($n = 48$) |
| (C) | Effective exercise time | 3344 | 3373 | 3198 | 3206 |
| | Average exercise HR | 124 | 125 | 127 | 124 |
| | Type of exercise | 1, 9 | 1, 4 | 1, 4 | 1, 4 |
| | Mets-min | 8918 | 10,900 | 8511 | 7211 |
| | Exercise frequency | 3.87 | 4.08 | 4.14 | 3.55 |

The suggested effective exercise time, average exercise HR, type of exercise, METs-min and exercise frequency of 2m-SEM in each group are shown in Table 3B. The required effective exercise time and recommended type of exercise in each month are basically same as 1m-SEM.

The suggested effective exercise time, average exercise HR, type of exercise, METs-min and exercise frequency of 3m-SEM in each group are shown in Table 3C. The required effective exercise time and recommended type of exercise in each month are similar to 1m-SEM and 2m-SEM as well.

### 3.3. Testing Accuracy of PV Modules

The mean square error (MAE) of 10-fold cross validation and final testing MAE of 1m-PV, 2m-PV and 3m-PV are shown in Table 2B. The value of MAE remains about 2.9 BPM in average during testing procedure. Except from the last layer of PV modules and SEM modules' neural network structure, the structure from the first layer to the third layer are the same. However, PV modules exists more difficulty to minimize the error than SEM modules.

## 4. Discussion

In this study, we constructed an exercise prescription system by real-world exercise data. However, the total amount of real-world data still remains lots of insufficient. Data oversampling and synthesis is applied to overcome the problem in this paper. Apart from this, the system has not been online to Joiisports platform for further testing yet. It will be a part of our future work. Such study is needed to further improve the structure, learning function, design, and also examine relations between exercise prescription and practical benefits. Eating preferences, drug history and other daily habits are not considered in this study. These variables will be added to our exercise prescription system for more precise recommendations.

The data with rest HR dropped more than 4 BPM after one-month, two-month and three-month exercise are the criterion to extract training data in the experiment. Recently, Joiisports developed a human body dash board able to collect five-dimension body information [26]. The user interface of the five-dimension human body dash board is illustrated in Figure 3. Those five dimensions including stamina, sleep, stress, vitals and shape. Stamina including rest HR, HR recovery rate and $VO_2$max. Rest HR and HR recovery rate are two of the indices to evaluate the ability of cardiac function. $VO_2$max refers to the ability of body oxygen delivery and usage efficiency. Sleep including sleep quality and blood oxygen saturation. Sleep quality is evaluated by different sleep stages. Those sleep stages including awake, rapid eye movemnet (REM) sleep and non-REM sleep. Besides, both light sleep and deep sleep are occurred in non-REM sleep. Stress including daily stress, awaken stress and heart rate variability (HRV). Standard deviation of normal to normal intervals (SDNN) calculated from electrocardiogram or photoplethysmogram data is one of the indices used to evaluate HRV [27]. Usually, high SDNN value refers to better cardiac ability. Reasonably, cardiac beating with high variations is confidential to adapt different external situation. Vitals including systolic blood pressure, blood sugar, cholesterol and uric acid. Shape including BMI, body fat percentage, waist and waist-to-hip ratio. In this study, rest HR is only included in one of the dimensions. Constructiing an exerisre prescription system based on the combinations of the above five-dimension body information is able to give user more solid and wide range of suggestions and analysis.

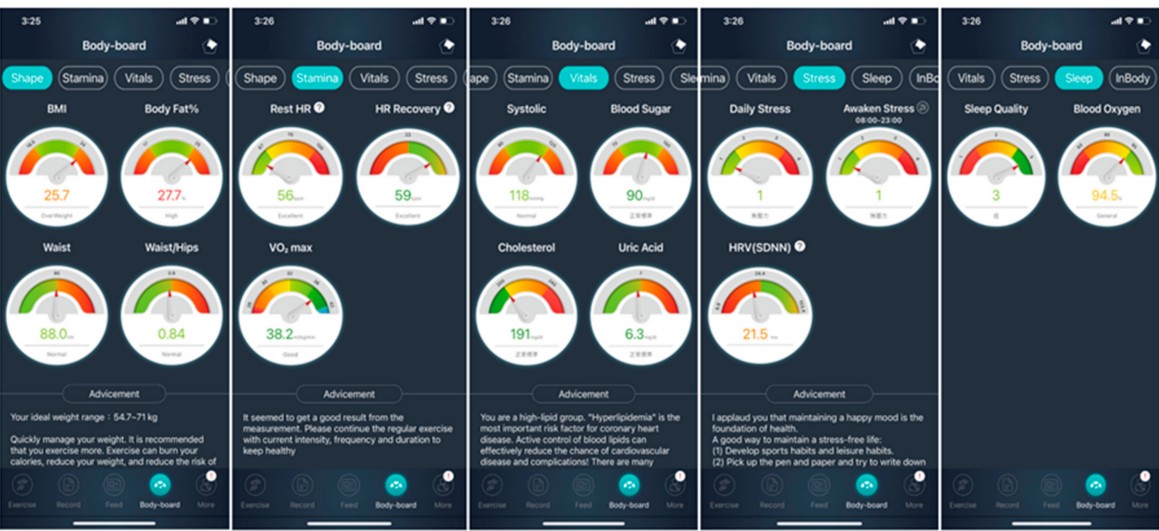

**Figure 3.** User interface of human dashboard from Joiisports.

METs-min is defined as MET value multipled by total exercise time (Minute). The World Health Organization (WHO) suggests that 600 METs-min is the minimum required within one-week exercise. However, research [28] found that for high-risk groups with diabetes, breast cancer, colon cancer and other diseases, exercising 3000–4000 METs-min per week is the most effective way to improve health. The suggested METs-min value in our study is range from 7000–12,000 per month. Converting into the required METs-min value in one week, it seems remains some insufficient compared to other research's suggested METs-min value. Despite this, exercise must be progressive. Intensity and duration time must be gradually increased according to personal physical fitness. On the other hand, equal amount of METs-min is able to achieve by both low intensity high duration exercise and high intensity low duration exercise. However, whether these two types of exercise bring in the same effect still needs to be explored.

Although the exercise prescription system for sub-health group has been developed in this study, whether people could stick on exercise still exits big challenge. It is more difficult for people to form exercise habit compared to develop a good exercise prescription system. How to

promote exercise has always been a topic that the world must work hard on. For the current enterprise, nearly 80% of employees always work overtime. The main reason for lacking of exercise is due to busy work. Therefore, if enterprise can promote and inspire employees exercise, and provide an exercise environment, this problem will be improved effectively.

## 5. Conclusions

In this study, a three-month exercise prescription system based on a large exercise platform dataset has been proposed for the sub-healthy group. The 1m-SEM, 2m-SEM and 3m-SEM modules obtained 95.8%, 100% and 95% accuracy of recommending the correct exercise mode for each user. The 1m-PV, 2m-PV and 3m-PV modules obtained 3.15, 2.89 and 2.75 BPM mean absolute error to predict the rest HR difference after following the 1m-SEM, 2m-SEM and 3m-SEM's recommendation. The system provides quantitative exercise prescriptions to guide sub-healthy group to engage in valid exercise programs and able to be implemented in smartphone application for future usage. Although the purposed system brings benefit in health-related improvement, evaluating personal health condition based solely on rest HR could limit the outcome. Constructing a comprehensive exercise prescription system with supporting multiple aspects can provide users with more integral and reliable suggestions. Besides, our system is focused on sub-healthy group. It is not applicable to other specific population. A robust system must require the capability to provide suggestions to different ethnic groups rather than the specific population. Integrating worldwide data will be a challenge in not only global health issue but also any health-related applications as well.

**Author Contributions:** Conceptualization, H.-K.C.; methodology, H.-K.C.; software, H.-K.C.; formal analysis, H.-K.C.; investigation, F.-H.C.; resources, F.-H.C.; writing-original draft preparation, H.-K.C.; writing-review a visualization, S.-F.L.; supervision, S.-F.L.; project administration, S.-F.L. All authors have read and agreed to the published version of the manuscript.

**Funding:** This project was not funded with any specific grant from any funding agency in the public, commercial or not-for-profit sectors.

**Institutional Review Board Statement:** The database be applied in this study is open access for employees and students of National Chiao Tung University, Hsinchu, Taiwan. The usage privilege of Joiisports database in this study obtained the written consent by JoiiUp Technology Corporation. Each user's name from database has been de-identified before starting the experiment.

**Informed Consent Statement:** Informed consent was obtained from all subjects involved in the study.

**Data Availability Statement:** Exercise data in this study are unavailable on sharing due to property of JoiiUp Technology Corporation.

**Acknowledgments:** The authors thank JoiiUp Technology Corporation from Hsinchu, Taiwan for providing the data on this study.

**Conflicts of Interest:** The authors declare no conflict of interest.

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
