# Peer review of "An AI-Based Exercise Prescription Recommendation System"

_applsci, doi:10.3390/app11062661_

Round 1

Reviewer 1 Report

The present work is an important contribution to the IT- and AI-based training recommendation in active leisure time athletes in sports. The approach is based on the special conditions and situation in Taiwan. A transfer to other countries with different infrastructures is not directly possible or questionable. Nevertheless, significant approaches for the use of AI in sports are shown.The study is well designed and well presented. The authors provide important data and results practical approach to AI in sports.

The authors should also be pointed out that similar exercise prescription for health (EPH) have been published elsewhere (www.efsma.eu and Sustainability). (see below).

Nonetheless from this compelling paper, some comments and questions remain.

What is the transferability to other sports systems ?
The present system has, after the description, no further self-learning function for further development.

A pre- participation evaluation (PPE) of athletes does not take place, for example, in order to recognize possible to the athletes in the sport, particularly in the elderly or very old person.

Admittedly,this is only given by analyzing a resting ECG in those athletes. So the question to the authors is, whether ths system can be enlarged to PPE?

It would be helpful for the reader if the abbreviations were listed and explained again separately in a box.

The recommendations are based on resting heart rate. Studies on this are listed as evidence. However, it is not explained, especially for the elderly, whether there is a drug influence (e.g. ß-blocker) folowing treatment in diseases.

The work also does not show to what extent a transfer to another institution is possible, or whether this is tied to the supporting company.

The requirements of the publisher are only quoted, but not answered.
Line: 280,288,295 (Institutional review... , informed consent...,Conflict of interest....)
must be filled in and answered.

References

Physical Activity, Exercise Prescription for Health and Home-Based Rehabilitation Sustainability 2020, 12, 10230;doi:10.3390/su122410230   www.efsma.eu (homepage)
Consider also
:Schwalbe N. et al.:Artificial intelligence and the future of global health. Lancet,395,2020:1579 ff

Author Response

Response to Reviewer 1 Comments

Point 1: The authors should also be pointed out that similar exercise prescription for health (EPH) have been published elsewhere (www.efsma.eu and Sustainability).

Response 1: We appreciate for reviewer’s valuable opinion and comments. We expanded the background information in the Introduction section to include the previous development of exercise prescription for health. (Line 24 to Line 30)

“The current AI-driven global health interventions ranges in four categories relevant to global health researchers: (1) diagnosis, (2) patient morbidity or mortality risk assessment, (3) disease outbreak prediction and surveillance, and (4) health policy and planning [9]. Focus on health policy and planning of previous healthcare recommendation systems, collaborative filtering, content-based, knowledge-based and hybrid approaches are the basic recommendation techniques in health recommender systems [10,11].”

Point 2: What is the transferability to other sports systems?

Response 2: The transferability of our study is the structure of the exercise prescription system based on rest heart rate. This approach provides other researchers with a new aspect to formulate exercise programs for different ethnic groups. Besides, the data pre-processing and labelling method is an alternative for researchers to construct different types of exercise prescription system to enhance final testing accuracy for recommending high precision exercise prescription. The system could also be applied to hospital, healthcare centres and other medical institution.  

Point 3: The present system has, after the description, no further self-learning function for further development.

Response 3: Thank you for this important suggestion. The exercise prescription system presented in our study is based on labelled data. Applying self-learning approach based on unlabelled data would be an important extension of our approach to build exercise prescription model. In our study, we applied labelled learning method to construct the system. Constructing exercise prescription system based on different learning methods will be experiment in our future work. (Line 239 to Line 241)

Point 4: A pre- participation evaluation (PPE) of athletes does not take place, for example, in order to recognize possible to the athletes in the sport, particularly in the elderly or very old person. Admittedly, this is only given by analysing a resting ECG in those athletes. So the question to the authors is, whether this system can be enlarged to PPE?

Response 4: Based on the data source of our system, the population is different with athletes. The exercise mode remains different between the “regular” population and athletes. Our system is not applicable for other specific population. The PPE system for athletes should be built based on athletes’ data. If we could acquire athletes’ personal and sports data, it is possible to construct their individual PPE model. (Line 300 to Line 303)

Point 5: The recommendations are based on resting heart rate. Studies on this are listed as evidence. However, it is not explained, especially for the elderly, whether there is a drug influence (e.g. ß-blocker) following treatment in diseases.

Response 5: Appreciate for reviewer’s valuable opinions. The information of daily habits, drug history, eating preferences of each person will be considered in our future work to improve our exercise prescription system for more precise recommendations.  (Line 241 to Line 243)

Point 6: The work also does not show to what extent a transfer to another institution is possible, or whether this is tied to the supporting company.

Response 6: Appreciate for reviewer’s comments. The system of our study is part of Joiiup Technology Corporation’s property. However, this result is available for other exercise platform for other applications.  

Point 7: The requirements of the publisher are only quoted, but not answered.
Line: 280,288,295 (Institutional review... , informed consent...,Conflict of interest....)
must be filled in and answered.

Response 7: Sorry for the mistakes. We have revised it from Line 309 to Line 319.

Reviewer 2 Report

This paper provides a guided exercise prescription system for subhealthy groups using exercise community data from Taiwanese JoiiSports to train an AI model. The system consists of six modules, including three-month suggested exercise mode, predicted value of rest heart rate difference after following three-month suggested exercise mode, two-month suggested exercise mode, predicted value of rest heart rate difference after following two-month suggested exercise mode, one-month suggested exercise mode and predicted value of rest heart rate difference after following one-month suggested exercise mode.

The paper seems to have been well planned and thought over. The authors gained access to real-world data from JoiiUp Technology Corporation and their Joiisports platform. This makes the results very interesting.

I like Figure 2 very much, which shows the overview of SEM modules’ training and testing procedure and the overview of PV 124 modules’ training and testing procedure. It is very legible. However, Figure 1 is not so clear, and I would be very glad if the authors could improve on that. I would also recommend explaining the SEM and PV shortcuts used a number of times, both in the main text at their first instances and in the Figure’s legend.

The literature review is good in some parts and somewhat scarce in others, lacking newest papers especially in the field of recommendation systems in general and neural networks. There is also some cited research which may be considered outdated as it is older than 10 years. It would be good to include in the Introduction a general introduction to recommendation systems, emphasizing their importance, their wide scope of usability, high value of monitoring data from real online platform users. Authors could refer there to recent international papers such as a paper published in Sensors journal titled Pharos 2.0—A physical assistant robot system improved; a paper published in Electronics journal titled Deep learning-enhanced framework for performance evaluation of a recommending interface with varied recommendation position and intensity based on eye-tracking equipment data processing; and a paper published in Procedia Computer Science titled Human-website interaction monitoring in recommender systems.

I would also recommend some work on the Conclusions section which should be elaborated on. This section should include the main conclusions, the limitations of your research, and I would be glad to see there future directions of your interesting research. Of course, the limitations may be a good starting point for further directions of your studies.  

The presentation has many errors, so there is some proof reading work necessary, too. Just to mention a few, a typo in the name of the Institute of Biomedical Engineering, lines 46-50 wrongly separated, missing articles (e.g. line 141 “due to unbalance problem”), wrong verbs (e.g. line 25 “remains higher risk”) etc. Some positions in the References have been formatted in a wrong way. What is more, some parts of the paper are not clearly worded and the sentences are too long for English language. The paper should be read by authors with caution and maybe the unclear parts could be rewritten.

Please consider the recommendations as friendly constructive feedback to improve the quality of your paper.

With best regards

Author Response

Response to Reviewer 2 Comments

Point 1: Figure 1 is not so clear, and I would be very glad if the authors could improve on that. I would also recommend explaining the SEM and PV shortcuts used a number of times, both in the main text at their first instances and in the Figure’s legend.

Response 1: Appreciate for reviewer’s opinion and comments. We have adjusted the resolution of Figure 1 (Page 3). The definitions of SEM and PV modules have been added to Line 62 to 64 and Line 142 to 144.   

Point 2: The literature review is good in some parts and somewhat scarce in others, lacking newest papers especially in the field of recommendation systems in general and neural networks. There is also some cited research which may be considered outdated as it is older than 10 years. It would be good to include in the Introduction a general introduction to recommendation systems, emphasizing their importance, their wide scope of usability, high value of monitoring data from real online platform users.

Response 2: Appreciate for reviewer’s opinion. We have added the following content to the Introduction section to introduce a general concept and importance of recommendation systems. The high value of real- world data has been described as well (Line 16 to Line 30)

“In the past few decades, lots of clinical data have been collected across different sites. With the growth of information technology, these data provided a high value of digital in-formation to integrate into the healthcare recommendation system. These systems gave patients a personalized recommendation and improved understanding of their medical condition. Personalized diets, exercise routines, medications, disease diagnoses, and other healthcare services all belong to the domain of healthcare recommendation systems. In addition to health-related recommendation systems, various recommendation systems have been integrated into online retailers, streaming services, social networks, physical assistants, and e-commerce applications [6-8]. The current AI-driven global health interventions ranges in four categories relevant to global health researchers: (1) diagnosis, (2) patient morbidity or mortality risk assessment, (3) disease outbreak prediction and surveillance, and (4) health policy and planning [9]. Focus on health policy and planning of previous healthcare recommendation systems, collaborative filtering, content-based, knowledge-based and hybrid approaches are the basic recommendation techniques in health recommender systems [10,11].”

Point 3: I would also recommend some work on the Conclusions section which should be elaborated on. This section should include the main conclusions, the limitations of your research, and I would be glad to see the future directions of your interesting research.

Response 3: Appreciate for reviewer’s opinion and comments. Several limitations of the current exercise prescription system have been added to the Conclusion section. (Line 296 to Line 304)

“Although the purposed system brings benefit in health-related improvement, evaluating personal health condition based on rest HR barely is not solid enough. Constructing a comprehensive exercise prescription system with supporting multiple aspects can pro-vide users with more integral and reliable suggestions. Besides, our system is focus on sub-healthy group. It is not applicable for other specific population. A robust system must require the capability to provide suggestions to different ethnic groups rather than the specific population. Integrating worldwide data will be a challenge in not only global health issue but also any applications as well.” 

Point 4: The presentation has many errors, so there is some proof reading work necessary, too. Just to mention a few, a typo in the name of the Institute of Biomedical Engineering, lines 46-50 wrongly separated, missing articles (e.g. line 141 “due to unbalance problem”), wrong verbs (e.g. line 25 “remains higher risk”) etc. Some positions in the References have been formatted in a wrong way. What is more, some parts of the paper are not clearly worded and the sentences are too long for English language. The paper should be read by authors with caution and maybe the unclear parts could be rewritten.

Response 4: Thanks a lot for the reviewer’s valuable opinion to enhance the quality of this study. We have revised the typo, missing articles, wrong verbs and positions of references of this study. The English writing has also been checked and corrected to the best of our knowledge.
